# History of Polish Canidae (Carnivora, Mammalia) and Their Biochronological Implications on the Eurasian Background

**DOI:** 10.3390/genes14030539

**Published:** 2023-02-21

**Authors:** Adrian Marciszak, Aleksandra Kropczyk, Wiktoria Gornig, Małgorzata Kot, Adam Nadachowski, Grzegorz Lipecki

**Affiliations:** 1Department of Paleozoology, University of Wrocław, Sienkiewicza 21, 50-335 Wrocław, Poland; 2Faculty of Archaeology, University of Warsaw, Krakowskie Przedmieście 26/28, 00-927 Warszawa, Poland; 3Institute of Systematics and Evolution of Animals, Polish Academy of Sciences, Sławkowska 17, 31-016 Kraków, Poland

**Keywords:** evolution, biochronology, paleoecology, competition, extinction

## Abstract

The remains of 12 canid species that date back ca. 4.9 myr have been found at 116 paleontological localities. Among these localities, eight are dated to the Pliocene age, 12 are dated to the Early Pleistocene age, 12 are from the Middle Pleistocene age, while the most numerous group includes 84 sites from the Late Pleistocene–Holocene age. Some, especially older forms such as *Eucyon odessanus* and *Nyctereutes donnezani*, have only been found at single sites, while the remains of species from the genus *Lycaon*, *Canis* and *Vulpes* have been recorded at numerous sites from the last 2 myr. Ancient canids such as *Eucyon* and *Nyctereutes* had already vanished from Poland in the Earliest Pleistocene, between 2.5 and 2.2 myr ago. Poland’s extant canid fauna is characterised by the presence of two new species, which spread into the territory due to a human introduction (*Nyctereutes procyonoides*) or natural expansion (*Canis aureus*). Research indicates a strong competition between dogs, especially between *Lycaon*, *Canis* and *Cuon*, with a strong lycaon-limiting effect on the wolf between 2.5 and 0.4 myr ago. After the extinction of *Lycaon lycaonoides*, *Canis lupus* evolved rapidly, increasing in number and size, and taking over the niche occupied by *Lycaon*. In order to reduce competition, the body size of *Cuon alpinus* gradually reduced, and it became an animal adapted to the forest, highland and mountain environments. Generally, the history of canids in Poland is similar to that known of Eurasia with some noteworthy events, such as the early occurrence of *Canis* cf. *etruscus* from Węże 2 (2.9–2.6 myr ago), *Lycaon falconeri* from Rębielice Królewskie 1A or one of the latest occurrences of *L. lycaonoides* from Draby 3 (430–370 kyr). Predominantly lowland or upland in the southern part and devoid of significant ecological barriers, Poland is also an important migration corridor in the East–West system. This 500–600 km wide corridor was the Asian gateway to Europe, from where species of an eastern origin penetrated the continent’s interior. In colder periods, it was in turn a region through which boreal species or those associated with the mammoth steppe retreated.

## 1. Introduction

A biostratigraphical and paleoecological analysis of Europe’s faunal changes would not be complete without including representatives of the Canidae family. Abundant and well-represented in the Plio–Pleistocene, they played an important ecological role among carnivoran guilds and were high-level competitors. Their appearance and dispersal were considered among the main bioevents and thus are important biochronological markers that can be correlated with main European faunal turnovers [1]. Among them, the most well-known is the “wolf-event”, used for the expansion of the wolves or at least wolf-like dogs into Europe ca. 2 myr [2,3,4,5]. Recently, it has been discovered that members of the genus *Canis* appeared in Europe much earlier, with the oldest occurrence from the French site Vialette, dated at 3.2–3.1 myr old [6,7,8,9]. This event is followed by high taxonomic diversity during the Early to Middle Pleistocene and, later, by a specific lineage with high size variation during the Middle and Late Pleistocene [1].

Similar to other widely distributed mammals, canids demonstrate a high morphological and size (clinal) variability. This is, however, often underestimated and leads to the creation of multiple taxa from a biogeographical basis. The body size of canids can be inferred from the size of anatomical elements such as teeth and bones. Canids’ geographical distributions are particularly wide across the earth. They cover Africa, both Americas, Eurasia and, as introduced, also Australia [1]. Such wide distributions cover several distinct climates and vegetation environments. This often implies a morphological, biometrical and genetic variability for the species, expressed as subspecies or geographical variants in taxonomy. The morphology and body size are among the two main analytical parameters of canids.

However, despite their remains being commonly found across Poland, contrary to other carnivores such as ursids and cats, canids as a group have never been an object of intensive, comprehensive study. The Polish territory has not usually been considered an area that yielded fossils of Canidae. The presence of different canids, mostly *C*. *lupus* remains, in a Polish territory had already been mentioned as of the 19th century [10,11,12,13,14,15,16,17]. However, apart from a few articles [18,19,20,21,22,23,24,25,26,27], Polish canids have only rarely been the subject of detailed scientific interest. These previously mentioned publications are purely descriptive and deal with faunal assemblages or isolated finds. Wolsan [28,29,30] was the first to publish comprehensive papers on all canids. These studies included a list of sites and plentiful descriptions of fossil remains. In these papers, however, there is no reference to any biochronological scheme and the lists of taxa are often incomplete. This study is the first comprehensive account of all known records of canids from Poland and the data presented are a result of an extensive revision. The work includes both quantitative and qualitative updates; however, it is neither exhaustive nor complete. It is very likely that there are still many finds that have not been reported to scientific institutions or museums, or else remain in private collections. They were often found by local reserchers and may have been sold on account of their great value and rarity. The material from some localities, e.g., caves of Połom or Węże 1, is scattered over many museums and private collections and it still contains hundreds of bones of different species to be further studied. This broad query and revision caused multiple new findings and descriptions of new taxa in Polish fossil theriofauna, e.g., *C. alpinus* [31].

This paper considers the importance of the morphology and the body size of canids from Poland on the European background. As a country, Poland is characterised by high regional peculiarities and regarded as an important, ecological corridor, especially in the east–west, geographical orientation. In the following paper, the canid remains from Poland were placed in a biochronological, European framework. The contribution of canids as biochronological taxa is also discussed.

## 2. Materials and Methods

The geologic timescale and subdivisions were based on the International Chronostratigraphic Chart (v2018/08) “http://www.stratigraphy.org/index.php/ics-chart-timescale (accessed on 6 Janurary 2023)” [32,33,34,35]. The biochronological division of Canidae is based on three main parameters: (1) the evolutionary stage, (2) the composition of the mammal assemblages in terms of genera and species and (3) the first appearance (FAD—First Appearance Datum) and last appearance (LAD—Last Appearance Datum) of the particular taxa. A δ18O curve is based on a benthic curve proposed by Lisiecki and Raymo [36]. A detailed list of Polish localities, with a list of species recorded at the site, age and full citation was given in Appendix A.

The nomenclatural codification follows the 4th edition of the International Code of Zoological Nomenclature (ICZN, 1999). Capital and lowercase letters, C/c (canines), I/i (incisors), P/p (premolars) and M/m (molars), refer to upper and lower teeth, respectively. The biochronology of the Mammal Ages was based on the macromammal assemblages [32,37,38,39,40,41,42,43,44,45,46,47,48,49,50,51].

## 3. Results

### 3.1. General Remarks

Canid remains were found in 116 Polish paleontological localities, dated ca. 4.9 myr (Figure 1; Table 1). Among them, 8 were dated to the Pliocene, 12 were dated to the Early Pleistocene, 12 were of the Middle Pleistocene age, while the most numerous group had 84 localities from the Late Pleistocene–Holocene. The presence of members of at least 12 species and 16 chronoforms was documented in Polish territories (Figure 1; Table 1). The obtained AMS dates directly confirmed the occurrence of *C*. *lupus*, *C*. *a*. *europaeus*, *V*. *vulpes* and *V*. *lagopus* in the chronological framework (Table 2).

The intense industrialisation of Western Poland in the second half of the 19th century and thus a significant intensification of earthworks resulted in an increase in the number of findings. These works included, among others, the railway and road infrastructure, the development of towns and villages, as well as a broad-based action of the drainage and regulation of water courses. Additionally, the growing interest in the archaeological research of caves and the exploitation of their sediments for producing fertilisers provided a significant amount of the material. Due to their accidental character, the exact location and stratigraphical context of those findings are usually scarce, if present at all. The accumulation of remains may be an outcome of natural or catastrophic death, slope or fluvial processes, activity of carnivores and some additional factors. The materials from the Polish open-air sites were most likely accumulated through natural processes. In addition, with the exception of Kraków-Spadzista, there is no evidence of human activity. A vast majority of the canid remains were found in localities concentrated in Southern Poland (Figure 1).

### 3.2. Neogene

The Polish records of Neogene canids are relatively rare and consist of a few specimens in isolated occurrences. The oldest known canids from Poland are the undetermined remains of a Canidae indet. of Podlesice and layers 4-5 of Mała Cave, dated on MN 14 (4.9–4.2 myr old) (Figure 1). In a few other sites such as Ewy Cave, Raciszyn 1 and Mokra 1, all dated at MN 15b (3.8–3.4 myr old), the remains probably belonging to *Nyctereutes* sp. have been found. Similarly dated, the only Polish occurrences of *Eucyon* cf. *odessanus* and *N. donnezani* were documented at Węże 1 (3.8–3.4 myr old) [20,21,22,23,25,26,27,28,52]. The previously reported presence of *Canis* at Węże 1 [26] was not confirmed [27]. The fauna located nearby at Węże 2 differs significantly from that of Węże 1 and includes species characteristics for the MN 16b, e.g., [53]. Taking into consideration the whole faunal assemblage, the latest Pliocene age (~2.9–2.6 myr ago) was proposed for this locality [28,54,55]. At Węże 2, the oldest and first occurrences of *Canis* cf. *etruscus* and *Vulpes* cf. *alopecoides* were found [56].

### 3.3. Quaternary

Since the earliest Pleistocene, the number of Polish records of canids has considerably risen. Ancient lineages from the genera *Eucyon* and *Nyctereutes* disappeared, while new large, social canids appeared. The oldest Polish record of the genus *Lycaon* is represented by sole M1 of considerable size of the *L. falconeri* from Rębielice Królewskie 1A (2.4–2.2 myr old). The known remains of *C*. *etruscus* and *V. alopecoides* have been found at only a few other, similarly dated (2.4–2.0 myr old) localities such as Zamkowa Dolna Cave (fauna A), Kielniki 3B, Kadzielnia and Mokra 2.

Approximately 2 Myr into the Polish fossil record, *V. praeglacialis* appeared with the oldest occurrence from Kadzielnia (2.2–1.8 myr old) and Kamyk (1.9–1.8 myr old) [28,29,30]. In localities dated between 1.9 and 1.7 myr ago, such as Kadzielnia, Kamyk, Kielniki 3B and Przymiłowice 3A, very little material (mostly isolated teeth) of *C*. *etrusus*, classified previously as *Canis* sp., has been discovered and mentioned [28,29,30]. The abundant material of canids has been found in Żabia Cave (1.7–1.5 myr old), represented by *L*. *falconeri*, *C*. *etrusus*, *V*. *praeglacialis*, and *Vulpes praecorsac* Kormos, 1932. The remains of both foxes, although in smaller numbers, have also been found in younger sites, dated at 1.5–1.2 myr old, such as Kielniki 1, Przymiłowice and Zalesiaki A. A few of them were also found dated on the late Early Pleistocene fauna C from Zamkowa Dolna Cave [28,29,30]. Between 1.7 and 1.5 myr ago, *C*. *etrusus* was replaced in the fossil record by *C. mosbachensis*, which makes it the oldest record of this species from Poland. Similarly, *L. lycaonoides* replaced *L*. *falconeri*, and the remains of both large canids have been found in Przymiłowice from Zalesiaki A and in fauna C from Zamkowa Dolna Cave.

The early Middle Pleistocene is less represented than the Early Pleistocene in Polish biochronology and only three sites have been documented for this period. From the most known, Kozi Grzbiet (800–700 Kyr), the remains of *L*. *lycaonoides*, *C*. *mosbachensis* and *V*. *praeglacialis* have been found [28,29,30,57]. The same canid composition was found in Południowa Cave, which is currently the only Sudeten locality of the Middle Pleistocene age [58]. It is also the site with the oldest record of *V*. *v*. *angustidens*. This chronosubspecies from ca. 700 kyr began being a constant member of Polish and European paleoguilds [57]. *C*. *mosbachensis* is also present in slightly younger (600–500 Kyr) localities such as Sitkówka and Rębielice Królewskie 2, which is also the last occurrence of *V*. *praeglacialis* in Poland.

From the biochronological viewpoint, what are especially valuable are two Polish sites documented from the late Middle Pleistocene. Canid remains, dated at ~550–450 Kyr from Tunel Wielki Cave, consist of *L*. *lycaonoides*, *C*. *mosbachensis*, *C. a. priscus* and *V. vulpes* ssp. and show not only the first appearance of dhole in Poland [59], but also the environment before and slightly during the massive glaciation of MIS 12. The second site, Draby 3 (MIS 11), is similarly intriguing. For *L*. *lycaonoides* and *C*. *mosbachensis* it is the latest and relict record not only in Poland, but also in the whole of Eurasia, especially for the *Lycaon*. After MIS 11, no sites with remains of *L*. *lycaonoides* are known elsewhere [60].

Dated at MIS 9, the oldest appearance of *C. l. lunellensis* in a slender and relatively small form was documented in the lowermost layers of Biśnik Cave. This first, true wolf was, however, larger and more robust than *C*. *mosbachensis* from Tunel Wielki Cave or Draby 3. In those layers, *C*. *a*. *priscus*, *V*. *vulpes* and *V. lagopus* also occurred [61]. The last is also the first Polish appearance and specimens from these layers are characterised by the presence of numerous, intermediate features that were still far away from those known in the Late Pleistocene individuals. From similarly dated (MIS 9) layers 1-V of Deszczowa Cave came a similar canid composition with some differences. Isolated M1 from layers V of Deszczowa Cave was classified as *C. a. fossilis* [31]. Small, gracile *C*. *l*. *lunellensis* and *V*. *vulpes* are morphologically very similar to those from contemporary layers of Biśnik Cave. The remains of a smaller fox strongly resembled *V*. *praeglacialis* [31,62,63,64], which may suggest an even older age.

From dated MIS 8 sediments of the Sudeten sites, Naciekowa and Wschodnia Cave, came the first record of large *C. lupus*, metrically and morphologically comparable with the extant, Polish grey wolf [65,66]. Both records are regarded as among the oldest of large-bodied wolves in Europe and probably documented an eastern immigration to Central Europe. On average, the Polish wolves from the period of MIS 7-5e, similar to those from the lowermost part of Wierzchowska Górna Cave, layers 18-15 of Nietoperzowa Cave, layers 18-14 of Biśnik Cave, layers 4-3 of Dziadowa Skała Cave or lowermost horizon of Komarowa Cave, are slightly smaller than the extant, Polish *C*. *lupus* [66].

The period of MIS 7-5e is also the time when the canid fauna finally formed the composition known today, including *C*. *lupus*, *V*. *vulpes* and *V. lagopus* [24,28,58,67,68], and rarely *C. a. europaeus*, whose presence was confirmed from a few Late Pleistocene sites [31]. These canids are common and permanent members of the Late Pleistocene, Polish palaeofaunas, and their presence is documented by numerous cave and open-air sites across the country [24,28,58,67,68]. Additionally, from a few sites such as Rogóżka Cave or layers 7-5 of Biśnik Cave, the presence of *Vulpes corsac* was confirmed for the first time in Poland. This adds a fifth canid species and this small fox was an eastern immigrant. Its presence in the fossil record is an indicator of open areas such as the steppe tundra.

Although the species composition is essentially unchanged in the Late Pleistocene, during the last 100 Kyr, there are marked fluctuations in body size, especially in the case of *C*. *lupus* [58,66,67,68]. Canids’ body size appears to have been profoundly influenced by the harsh environmental conditions that were present, with a larger size potentially being a Bergman response to the severely cold climate. This, in combination with the strong competition with other larger carnivores, was very likely responsible for the substantial increase in size of *C*. *lupus*. Alongside the normal-sized individuals, dimensionally comparable with the extant Polish *C*. *lupus*, such as those from Cave no 4 on Mount Birów, Raj Cave, Kraków-Spadzista or Borsuka Cave [24,58,67,68,69], particularly large and robust individuals commonly appeared. In some, especially Sudeten (Radochowska, Obok Wschodniej Cave, Niedźwiedzia Cave) and those from Kraków-Częstochowa Upland such as the layers 7-5 from Biśnik Cave localities, whole populations were significantly larger than the extant *C*. *lupus* [58,66,67]. Maybe they were not so common, but these huge wolves have been found in some open-air sites, located in Polish lowlands [67,70,71,72]. The presence of these great wolves, referred to as *C. l. spelaeus*, is traditionally linked with the cooler phases of the Late Pleistocene. Their occurrence from multiple Polish sites is documented until the end of MIS 2.

Since MIS 3, the occurrence of *C*. *lupus*, *C*. *a*. *europaeus*, *V*. *vulpes* and *V*. *lagopus* is additionally directly confirmed by a series of AMS dates (Table 2). Among them, the dated 44.3-43 call Kyr record of *C*. *a*. *europaeus* from Radochowska Cave [31] or the occurrence of the huge *C*. *l*. *spelaeus* in Niedźwiedzia Cave during the cold period 31.2–30.2 call Kyr are especially noteworthy (Table 2). They also indicate that the cold-adapted *V*. *lagopus* survived until the late part of MIS 2 (Table 2) and the last known specimens dated ~13.8–13.7 call Kyr [63,73,74]. It probably retreated to the east and north from the Polish territory during the Early Holocene. From MIS 1, the remains of *C*. *l*. *spelaeus* and *V*. *corsac* are also already unknown. In general, like numerous other carnivores, *C*. *lupus* and *V*. *vulpes* cannot be grouped in a typical ‘Pleistocene’ or ‘Holocene’ faunal community. The extreme, ecological adaptability of both canids means that, in their case, we cannot talk about the recolonisation of the Polish territory after the retreat of the ice sheet. They just continued their presence in Poland, maybe only increasing in density due to the warming of the climate and expansion of various, preferable habitats.

For almost the whole Holocene, *C*. *lupus* and *V*. *vulpes* were at the core of canid fauna in Poland [24,28]. *C. l. familiaris* joined these species (Linnaeus, 1758). Their remains from Poland are not older than the Mesolithic, and shortly after, domestic dogs started to be the most commonly recorded canid from the archaeozoological sites [75,76]. Holocene records of *C*. *lupus* and *V*. *vulpes* are also common and widespread, and their presence has been confirmed from numerous cave and open-air sites [24,58,67,68,69] as well as from archaeozoological localities [75]. Finally, in the last 70 years, two new species of canids have appeared in Poland. The extant *N. procyonoides* (Gray, 1834) came to Poland in 1955 from areas of Belarus, Lithuania and Ukraine. In Finland, they were recorded for the first time in 1935, in Sweden in 1945–46, Romania in 1952, Slovakia in 1959, Germany and Hungary in 1961–62 and Norway in 1983. The first specimens were observed in and around the Białowieża Primeval Forest (NE Poland) and Hrubieszów in the Zamość region (SE Poland) [77,78,79,80,81,82]. *N*. *procyonoides* spread into the Polish territory in the NE to SW direction. Until the 1980s, their geographical range covered almost the entire area to the east of the Wisła river. By the early 1990s, the species had been recorded in almost the whole of Poland. The expansion was the result of finding a free ecological niche with favourable biotope conditions [82,83]. In 2015, in Poland for the first time appeared the golden jackal *C. aureus* Linnaeus, 1758 [84]. The records suggest a natural expansion into Poland, probably from different source populations [84].

**Table 2 genes-14-00539-t002:** AMS radiocarbon dating results of canids from Polish sites. All new dates were calibrated using the programme IntCal 20 according to [85]. Within the text, only calibrated data were used.

Species	Site	Bone	Lab. Code	%C	%N	^14^C Age	Cal. BP 95.4%	Source
*C. a. europaeus*	Radochowska Cave	maxilla	MAMS-48222 *	45.7		40.480 ± 380	44.310–42.960	[31]
*V. vulpes*	Obłazowa 2	mandible	OxA-3696			33.430 ± 1230	38.885–37,675	[86]
*V. vulpes*	Cave 4 on Mt Birów	mandible	Poz-27279	9.9	3.5	27.980 ± 220	32.048–31.681	[69]
*C. l. spelaeus*	Niedźwiedzia Cave	costae	Poz-143821	10.0	1.8	26.640 ± 290	31.078–30.858	new
*V. lagopus*	Deszczowa Cave, l. VIIa	radius	Poz-26126	8.8	2.9	24.620 ± 200	29.038–28.736	[63]
*V. vulpes*	Deszczowa Cave, l. VIIa	humerus	Poz-23437	7.3	1.5	24.470 ± 260	28.824–28.628	[87]
*V. lagopus*	Kraków-Spadzista	maxilla	Poz-28734	8.0	2.8	24.640 ± 160	29.061–28.753	[88]
*V. lagopus*	Obłazowa Cave, l. 7	femur	Poz-22686	13.1	3.0	23.500 ± 230	27.827–27.627	[87]
*V. lagopus*	Kraków-Spadzista	mandible	Poz-28733	6.6	1.9	23.150 ± 190	27.590–27.274	[88]
*V. lagopus*	Deszczowa Cave, l. VIIIa	humerus	Poz-25075	10.2	3.3	22.530 ± 160	27.091–26.868	[63]
*V. vulpes*	Deszczowa Cave, l. VIIIa	tibia	Poz-22666	7.0	1.2	20.280 ± 130	24.535–24,188	[87]
*V. lagopus*	Kraków-Spadzista	mandible	Poz-51377 ***			178.70 ± 110	21.906–21,456	[89]
*V. lagopus*	Jasna Strzegowska Cave	mandible	Poz-24205	16.7	5.1	14.400 ± 80	17.784–17,375	new
*V. lagopus*	Nad Potoczkiem Cave	mandible	Poz-23646	16.6	4.2	13.560 ± 70	16.487–16,267	new
*V. lagopus*	Wilczyce	Tooth **	OxA-16728			13.180 ± 60	15.928–15,710	[74]
*V. lagopus*	Krucza Skała Shelter, l. 2-4	mandible	Poz-27245	7.1	2.1	12.970 ± 60	15.634–15,364	[63]
*V. lagopus*	Krucza Skała Shelter, l. 1	mandible	Poz-27246	7.7	2.2	12.920 ± 60	15.577–15,316	[73]
*V. lagopus*	Cave 4 on Mt Birów	mandible	Poz-27244	10.3	3.7	12.590 ± 60	15.114–14,926	[69]
*V. lagopus*	Puchacza Skała Shelter	mandible	Poz-27260	9.5	2.4	123.00 ± 70	14.328–14,132	new
*V. lagopus*	Wilczyce	bone	Ua-15723			11.890 ± 105	13.797–13,747	[74]

* in Marciszak [31] under the working number REVA-3413. ** perforated arctic fox tooth, pendant. *** The possibility of some contamination affecting the results cannot be excluded.

## 4. Discussion

### 4.1. Large Canids: Genus Eucyon, Lycaon, Canis and Cuon

The earliest advanced member of the tribe Canini Fischer, 1817 is the genus *Eucyon*. Its origin is identified in North America [90], where it appeared in the fossil record ca. 12 myr [90,91]. During a westward dispersal in the Miocene, the *Eucyon* species appeared in Central Asia [92], Europe [93,94,95] and Africa [96,97]. This was not a single event and during the late Miocene, a number of taxa migrated between North America and Africa and Eurasia [93,94,95,98,99,100].

*Eucyon cipio* (Crusafont Pairó, 1950) was the first canine from Eurasia, known from the dated MN 12 (~7.9–6.9 myr ago) Iberian sites, Concud and Los Mansuetos [93,94,95,101,102,103,104,105]. The dental morphology is primitive and suggestive of a taxon at an early stage of evolution before the differentiation of the genus *Canis*. *Eucyon* gave rise to the first representative of *Canis* ~6.5–6.0 myr ago in North America and did not survive into the late Miocene [91,95,106]. Contrary to that, in the Pliocene in Eurasia, the genus reached a relatively high diversity. In Europe its latest occurrence was documented from ~3.8–3.4 myr sites, Saint-Estève (France), La Calera 1 (Spain) and Red Crag (England) [93,94,95,102,105,107,108]. The material from these sites represents the jackal-sized *Eucyon adoxus* (Martin, 1973). The Eastern European, more or less, contemporary records from Moldova, Ukraine, Greece and Turkey were assigned to *E. odessanus* (Odintzov, 1967) [94,95,109]. To this group, a new Polish record from Węże 1 has recently been added, which is among the latest *Eucyon* occurrences. It still survived until the latest Pliocene (~2.8–2.7 myr ago) in Central Asia, Morocco and Turkey [92,95,110].

The extinction of the *Eucyons* in the latest Pliocene is correlated in time with the appearance of the first representatives of the genus *Canis* and *Lycaon* in Eurasia. Firstly, they appeared in Central Asia 3.4–3.3 myr ago (Figure 2) [60,94], while the earliest European record from the French site Vialette is dated at 3.2–3.1 myr old [6,7,8,9]. This implies that these wolf-sized forms appeared in Eurasia practically simultaneously in Asia and Europe. The Hungarian locality Osztramos 8 is slightly younger (2.8–2.6 myr old) [106]. It is known that *Eucyon* gave rise to *Canis* 6.5–6.0 myr ago in North America [95,101,106]. This scenario in Europe, like the evolution in situ, also cannot be entirely ruled out. The westward invasion of *Canis* dogs put an end to domination, especially as they emerged in a larger, social and much more advanced form. Here, it can be also added that any of the so-called “jackal” lineages such as *Canis arnensis* Del Campana, 1913 (and all related forms) never reached Polish territory. They were widespread across Southern Europe during the late Villafranchian and were replaced by *C*. *mosbachensis* 1.6–1.5 myr ago [111].

Shortly after the early wolves in Europe appeared, canids from the genus *Lycaon*, dominated ecosystems for the next 2 myr. Canids from the genus *Lycaon* appeared in Europe ~2.6 myr ago in a robust build and great-sized form, which did not particularly change for the next 2 myr [112]. Admittedly, their ancestor, *Sinicuon dubius* (Teilhard de Chardin, 1940), from the Tibetan region Zanda Basin dated at 3.8–3.4 myr ago, was the size of a large dhole *C. alpinus* (L m1 = 22.5 mm) [112,113,114,115,116,117,118]; their descendants were much larger and were similar in size to or exceeded the largest extant wolves [31]. The individuals from the oldest European record of Perrier-Roccaneyra (France; 2.6–2.5 myr old) already had the morphology and size characteristics of the Early Pleistocene *Lycaones* [112]. As shown in the analysis of the size variation over time, the dimensions of the *Lycaon* underwent only slight changes. It was a member of a quite stable paleoguild of ancient carnivores, including *C. etruscus*, *Ursus etruscus* Cuvier, 1823, *Homotherium latidens* (Owen, 1846), *Panthera gombaszoegensis* (Kretzoi, 1938), *Acinonyx pardinensis* (Croizet & Joubert, 1828), *Puma pardoides* (Owen, 1846), *Pachycrocuta brevirostris* (Gervais, 1850), *Chasmaporthetes lunensis lunensis* (Del Campana, 1914) and *Pliocrocuta perrieri* (Croizet and Jobert, 1828) [119,120,121,122]. Among them, *L*. *falconeri* held a very high position and, when grouped, was regarded among the dominant carnivore species [119,120,121,122].

Known from the late Pliocene, the relatively long-lived and seemingly fairly stable European carnivore guild during the Early Pleistocene was changing. *P*. *brevirostris* replaced *P*. *perrieri* and larger canids replaced *Ch*. *lunensis* [122,123,124,125]. In the habitats previously dominated by large hyenas, the social, highly intelligent, easily adaptable and aggressive canids from the genuses *Canis* and *Lycaon* emerged. They dominated these ecological niches for the next 2 myr. *L*. *falconeri* and the latter *L*. *lycaonoides* were as large or larger than the extant largest *C*. *lupus* [31,60]. Their great size and social behaviour were advantageous in the competition with *Canis* and *Ch*. *lunensis*. As an active social hunter, *Ch*. *lunensis* suffered mainly from the retrieval of prey, just as in extant times *L. pictus* often lost its prey to *Cr*. *crocuta*. Although *Ch*. *lunensis* had no counterpart among hyenas in extant faunas and is closest to *L*. *pictus*, many analogies can be found. In sufficiently numerous packs, *L*. *pictus* coped well in competition with *Cr*. *crocuta* and *Parahyaena brunnea* (Thunberg, 1820). Even the extant, relatively small packs of *L. pictus* were able to successfully compete in the habitats dominated by *Cr*. *crocuta*. However, as noted by Eaton [126], these data originate from recent populations in which the pack size of *Lycaones* appears to have been much smaller than in the early part of this century and before. Thus, historically, wild dogs were probably dominant to spotted hyenas.

*Ch*. *lunensis* was comparable in size to the modern *Cr*. *crocuta* but was much more lightly built, probably weighing at least 30% less. As research shows, a massive body structure is one of the factors that guarantees an advantage in aggressive encounters with other carnivorans of a similar size [127,128]. Examples include the *Panthera spelaea fossilis* (von Reichenau, 1906) and *H*. *latidens* [31,129], *P*. *gombaszoegensis* and *A*. *pardinensis* in the Pleistocene or the extant *C*. *lupus* and *Hyaena hyaena* (Linnaeus, 1758). We suggest that *Ch*. *lunensis* was not powerful enough to successfully compete with the pair of effective cursorial hunters such as *L*. *falconeri* and *C*. *etruscus*. As a result, this hyena disappeared as one of the first large carnivores ca. 1.8–1.6 myr ago. Contrary to *Ch*. *lunensis*, immense and extremely robust *P*. *brevirostris* was powerful enough to successfully compete not only with those canids, but also with *H*. *latidens* and *P*. *gombaszoegensis* [130,131,132].

A more marked increase in body size is noticeable from the Middle Pleistocene (ca. 0.8 myr ago) as a possible response to the increasing competition from the side of two great newcomers of African origin. Their appearance led to a complete reconstruction of the fauna of carnivores. The first to appear was *Crocuta crocuta* ssp., which quickly led to the extinction of *P*. *brevirostris* [31,121,123,130,131,132]. Shortly afterwards, the huge *P. s. fossilis* (von Reichenau, 1906) appeared, which led to a drastic decline in numbers and almost the complete disappearance of *H*. *latidens* and *P*. *gombaszoegensis* [64,133,134].

The Polish records from the genus *Canis* and *Lycaon* are slightly younger than the oldest European occurrences but still showed an early appearance of these canids (Figure 2). The oldest *Canis* is known from Węże 2 (2.9–2.6 myr ago), while the oldest *Lycaon* was found in Rębielice Królewskie 1A (2.6–2.2 myr ago). For the next 2 myr, the presence of the pair of two large canids in the fossil records, the larger *Lycaon* and smaller wolf, has been well documented across Eurasia by a number of localities [31,60,112,135,136,137,138,139,140,141,142,143,144,145,146,147,148,149,150,151,152,153,154,155,156,157,158,159,160,161,162,163,164,165,166,167,168,169,170]. Of these two, the long-legged *Lycaon* was much larger than the wolf and dominated the open grasslands. It had short, broad jaws and massive dentition, ideal for crushing and slicing [159]. In its lifestyle and hunting behaviour, *L*. *lycaonoides* was similar to the extant *Lycaon*. Long, slender legs allowed it long and persistent chases, due to which the tired prey was killed by evisceration [60,153,156,159,166]. The larger size of *L*. *lycaonoides* and its more robust build allowed it to hunt much larger prey. They also positively influenced the competition with other carnivores [153]. This canid occupied a particular area of the morphospace that is unlike any extant species [112]. It was capable of bringing down very large prey. The carcass would have been eaten whole, thanks to the sturdy dentognathic features.

The mighty *L*. *lycaonoides* was accompanied by the small *C*. *mosbachensis*. The slender build and small size of this canid were almost identical to the extant Arabian or Indian wolf [60,146,147,153,156,169,171]. It was probably more flexible ecologically and better adapted to changing conditions than the Eurasian *Lycaon*. By about 1.5 myr ago, members of the genus *Canis* lived in the shadow of the dominant *L*. *lycaonoides* (Figure 2). Relationships between these two canids were probably similar to the extant relationship between *C*. *lupus* and *Canis latrans* Say, 1823 in North America [172].

During the latest Early Pleistocene, robust *C. a. priscus* joined *L*. *lycaonoides* and *C*. *mosbachensis* (Thenius, 1954). Being of an intermediate size between *L*. *lycaonoides* and *C*. *mosbachensis*, the massive and short-legged dhole was adapted to life in the mountains, forest and upland areas. Being smaller than most of the larger carnivores, its in-group cooperation compensated for its lack of size. Extant dholes lived in packs of up to several dozen individuals [173]. *C*. *a*. *priscus* found favourable habitats, as evidenced by the increasing number of sites at which its remains have been found over time [174]. *L*. *lycaonoides* dominated the open lands, while *C*. *a*. *priscus* tended to prefer forests, mountains and highlands. *C*. *mosbachensis* coexisted in all these environments, trying to minimise competition with other canids by avoiding them or living in habitats less suited to them [171]. Today, the dhole and the small Indian wolf coexist quite well in Southeast Asia. The cases of aggression between them are rare [175]. *C*. *a*. *priscus* and *C*. *mosbachensis*, which had similar body sizes, probably had similar relationships. Relationships between the Eurasian *Lycaon* and the primitive dhole are more difficult to reconstruct, even if it has been suggested that *L*. *lycaonoides* was outcompeted by a primitive dhole in some isolated areas such as Java [176,177]. *L*. *lycaonoides* probably had a negative impact on the wolf population. The much more numerous but smaller *C*. *mosbachensis* could be actively killed and driven away from carcasses by a larger *Lycaon*, as larger canids usually dominated smaller species [178].

The earliest appearance of *C*. *a*. *priscus* is dated to the mid-Middle Pleistocene ca. 600–550 kyr ago [146,148,179,180]. Recent revisions show that material morphologically similar to large, robust dholes is also present in a few older European sites dated between 0.9 and 0.7 myr ago. *Cuon* has an Asian origin [118,146,181,182,183,184,185,186,187]. Since the genus has a common ancestor with *Lycaon*, the similarity between the Middle Pleistocene *Cuon* and *Lycaon* is sometimes so close that their bones have often been confused in the past. The largest individuals of *C*. *a*. *priscus*, such as those from Mosbach 2 or Petralona Cave, metrically matched with the smallest specimens of *L*. *lycaonoides* from Vallparadis [169].

Up to MIS 12, *L*. *lycaonoides* was widespread across Eurasia, but in a younger period (MIS 11), its occurrence was noted only rarely [60]. The only undisputed occurrences of *L*. *lycaonoides* from that time are the Polish site Draby 3 and (probably) the Hungarian site Vertesszöllös 2 [188]. Most likely during MIS 12, the combination of the influences of the harsh climate, changes in the ungulate fauna and increasing competition with *Cr*. *spelaea* and *P*. *s*. *fossilis* resulted in a drastic reduction in the number and range of *L*. *lycaonoides*. It was probably a similar process to what was noted in the 19th and 20th centuries for *L*. *pictus* in sub-Saharan Africa [189,190,191,192,193,194,195,196,197,198]. The balance between carnivores was disturbed, and a certain critical point in the *L*. *lycaonoides*–*C*. *mosbachensis* relationship was exceeded. Between 450 and 400 kyr ago, the Eurasian *Lycaon* was already too rare to be a real competitor and a limiting factor for the wolf. As soon as *L*. *lycaonoides* disappeared in Eurasia, a distinct increase in the number of sites and size of the wolf was observed (Figure 3; 1, 60, 153, 198). The obtained results of the estimated wolf body weight for a wolf from 2.0–0.5 myr ago show a significant similarity, regardless of age or geographic distances. The resulting estimates between different populations from that period are remarkably similar to one another. Taking into consideration their difference in age as well as regional distances, the apparent temporal and regional stability in size during the Early and Middle Pleistocene is notable [199,200].

In this matter, the two Polish records from Tunel Wielki Cave and Draby 3 are especially noteworthy. The first, from Tunel Wielki Cave, dated at MIS 13-12, documented the time where all three large canids, *L*. *lycaonoides*, *C*. *a*. *priscus* and *C*. *mosbachensis*, still occurred together spatially and temporally [59]. It is also the last time when *Lycaones* was still widespread across Europe and before the massive glacial age of MIS 12, which was critical and crucial for the disappearance of this species in Eurasia [31]. In turn, Draby 3 showed the world already after this massive glaciation a strongly rebuilt fauna. This record represents the latest, relict occurrence of *L*. *lycaonoides*. The rich faunal assemblage of Tunel Wielki Cave is characteristic of mixed open grassland, closed woodland and aquatic environments based on the presence of numerous grazer animals. The presence of numerous thermophilus taxa indicate warmer-than-present interglacial conditions [59,201]. The assemblages from the younger Draby are also characterised by temperate conditions. The assemblage suggests the presence of a temperate climatic episode within a single complex interglacial period (Figure 3).

The climatic complexity of the Middle Pleistocene of Poland was characterised by multiple temperate episodes. Even though the assemblages investigated here represent the overall temperate interglacial conditions, the time interval was characterised by notable climatic upheavals. Following the Mid-Pleistocene Revolution (~1.2 myr ago), Northern Europe began to experience the first major land-based glaciations, with 100 kyr eccentricity-dominated glacial-interglacial cycles fully established ca. 800 kyr ago [201]. A notably cold period of MIS 22 was the first major cold event that led to substantial continental ice volumes equivalent to the later Pleistocene glaciations (e.g., MIS 16, 12, 6 and 4-2) [201,202]. However, periodical climatic fluctuations seem to have had a minimal effect on the *C*. *mosbachensis* body size over time. A slight size variation between different populations may have arisen from regional differences in the climate and environment. Temporal stability in the *C*. *mosbachensis* body mass occurred in concert with palaeodietary stability in the same Pleistocene populations [199,200,203].

The period between 400 and 300 kyr ago was a time when the dhole and the wolf were still close in body size, but the situation changed. The wolf slowly increased in size, reaching the sizes known today, and the number of individuals in packs grew to several dozen [1,144,198,204]. The wolf became the dominant canid in Eurasian ecosystems and took over a niche occupied so far by the *Lycaon* [31,171]. It was also the time when the first true wolves *C*. *l*. *lunellensis* appeared [1,144,186,198]. This process occurred both as the evolution in situ from *C*. *mosbachensis* and also as a replacement by these new wolves [205]. The process has most probably an irregular character. From contemporary, European localities, the remains of wolves were determined once as *C*. *mosbachensis* and once as *C*. *lupus* [154,165,181,182,186,188,206,207,208]. Some authors favoured the hypothesis that it was the replacement rather than the evolution in situ, especially when considering the large turnover recorded during and after the MIS 12 [205]. According to them, glacial conditions of MIS 12 may have played a role in triggering the spread of *C*. *lupus* into Europe. In this scenario, *C*. *lupus* may have dispersed into Europe, encountering or even interbreeding with *C*. *mosbachensis* which previously inhabited the region [209,210,211,212]. The timespan of MIS 11-10 as the time of interchange from *C*. *mosbachensis* into *C*. *lupus* is also documented in Polish territory. The individuals of *C*. *mosbachensis* from Tunel Wielki Cave (MIS 13-12) and especially those from Draby 3 (MIS 11) are large and robust, and their size places them among the largest known specimens. Metrically and morphologically similar populations are known from Vertesszöllös 2 (Hungary, MIS 11) [154,188] or layers 8-5 of the Kudaro 1 Cave (Georgia, MIS 11) [186].

Stratigraphically younger layers 19ad-19 of Biśnik Cave (MIS 10-9) documented the oldest presence of *C*. *l*. *lunellensis* in Poland, together with large *C*. *a*. *priscus* and a lack of *L*. *lycaonoides* (Figure 3). This was already the time when the Eurasian *Lycaones* no longer existed in Eurasia. From this period an increasing size process started. This is well documented by the wolf material from the Sudeten sites Naciekowa and Wschodnia Caves (MIS 8) [66]. The remains of *C*. *lupus* from MIS 7-5e, commonly found across Europe, are still slightly smaller than the average extant specimens. A recent analysis of cranio-dental measurements showed that *C*. *lupus* from the period MIS 7-5e was less adapted for fast flesh slicing and more adapted for the crushing of non-flesh foods, combined with comparatively weaker jaws than its later (MIS 5d-2) counterparts [199,200,203]. Contrary to that, the great Late Pleistocene *C*. *l*. *spelaeus* showed a changed skeletal development due to a preference for larger prey. It resulted in larger and more robust wolves with a shortened rostrum, the pronounced development of the temporalis muscles, proportionally enlarged and wider premolars and carnassials. These features were specialised adaptations for the processing of fast-freezing carcasses, especially during cooler phases, and associated with the hunting and scavenging of larger prey. Additionally, the body dimensions of *C*. *lupus* appear to have been profoundly influenced by harsh climatic and environmental conditions [199,200]. The association of a cold, low and relatively stable diversity of mammalian fauna during the coolest interstadials of the Late Pleistocene and great *C*. *l*. *spelaeus* was documented from multiple Polish sites. The robust build, large dimensions and cranio-dental adaptations at this time would have enabled these wolves to hunt larger prey than their MIS 7-5e or MIS 1 counterparts, making even adult *Bison priscus* Bojanus, 1827 a more attainable target [199,200].

Compared with the extant *C*. *lupus*, some Late Pleistocene populations of *C*. *l*. *spelaeus* showed an increase in tooth breakage. When the frequency and location of teeth fractures found in these wolves were compared with the extant *C. crocuta* (Erxleben, 1777), it was stated clearly that they were habitual bone crackers [203]. A genetic analysis confirmed that these ancient populations of *C*. *l*. *spelaeus* carried mitochondrial lineages which could not be found among the extant *C*. *lupus*, which implies their extinction [213,214]. This also indicates that *C*. *l*. *familiaris* dogs are descendents from those extinct lineages of *C*. *lupus* [215,216,217]. In Central Europe, the earliest *C*. *l*. *familiaris* remains, dated at 14.4–14.2 call kyr ago, were discovered in Bonn-Oberkassel [218,219,220]. Multiple pieces of evidence, such as an archaeological context, isotopic and genetic analysis, and morphological values, clearly indicate a domestic dog. Contrary to *C*. *l*. *familiaris* from Bonn-Oberkassel, the remains of so called “first dogs” from Moravian sites such as Dolní Věstonice, Pavlov and Předmostí, e.g., [221,222], were in fact more than highly variable populations of *C*. *l*. *spelaeus*. In this term, we agree with other authors such as Wilczyński [221], especially because our studies also supported this opinion. However, because the remains of *C*. *l*. *familiaris* from Poland are not older than the Mesolithic period [76], we did not reopen the discussion about the early history of the domestication.

### 4.2. Small Canids: Genus Nyctereutes and Vulpes

The origin of the genus *Nyctereutes* is highly debatable and rather dubious [223]. The oldest Eurasian occurrence is dated at ca. 4.9–4.2 myr ago and is represented by *Nyctereutes tingi* Tedford and Qiu 1991 from the Yushe Basin [223]. The first appearance in Europe is slightly younger (4.2–4.0 myr ago), with mainly documented presences of *N. donnezani* (Depéret, 1890) in sites such as French Perpignan or Spanish La Gloria 4 [105,224,225]. A bit later (4.0–3.9 myr ago), *Nyctereutes megamastoides* appeared (Pomel, 1842) with the first record from the Spanish locality Layna [223,226,227]. Canids from the genus *Nyctereutes* are characteristic members of the Pliocene and earliest Pleistocene Eurasian palaeofaunas (4.9–2.0 myr), and survived until the late Villafranchian. Eurasian sites younger than 2 myr ago documented the fast declining process of their diversity and range, most likely due to the competition with species from the genus *Vulpes*, until their final extinction between 1.8 and 1.6 myr ago [223,226,227,228].

Previously, it was agreed that *N*. *megamastoides* evolved to be larger and with stronger dentition from the primitive *N*. *donnezani* [52]. The wide geographical range, complicated evolutionary history and previous lack of clear, diagnostic features caused material in the past from the same site to be differently determined [52,229,230,231,232,233,234,235]. The same was true for the material of *Nyctereutes* from Węże 1 (3.8–3.4 myr), which is so far the only Polish record of this genus. First, it was described as *Nyctereutes* sp. [20,22], then classified once as *Nyctereutes sinensis* (Schlosser, 1903) [25], once as *N*. *donnezani* [52] and as *N*. *megamastoides* [25]. Finally, however, it was agreed to determine *Nyctereutes* from Węże 1 as *N*. aff. *donnezani* [28]. Additionally, our preliminary observations, made during the revision and study of the new material from this locality, confirmed this assignation. The only other Polish record of *Nyctereutes* came from the nearby and slightly younger (2.9–2.6 myr ago) site Węże 2, where a partially preserved hemimandible with p4-m1 had been found. After a consistent hiatus in the Polish history of *Nyctereutes* during the Pleistocene [226], *N*. *procyonoides* appeared in 1955 and spread into the territory from the NE to SW direction [77,78,79,80,81]. By the early 1990s, the species was already recorded in almost the whole of Poland [83].

Traditionally, three species of the genus *Vulpes* were recognised from the European Plio-Pleistocene: *V. alopecoides* (Del Campana, 1913), *V. praeglacialis* (Kormos, 1932) and *V. praecorsac* Kormos, 1932 [236,237]. The late Pliocene and earliest Pleistocene fox remains have been ascribed to *V*. *alopecoides*. The Early to Middle Pleistocene fossils have been determined mostly as *V*. *praeglacialis* and, but in much rarer cases, as smaller *V*. *praecorsac* on the basis of size. In the broad, most recent revision, all Early Pleistocene individuals were allocated to *V*. *alopecoides* and synonymising with the latter multiple species distinguished by previous authors [144,235,236,238,239,240]. The earliest European fox records are those from the Bulgarian site Musselievo [241,242,243] and Odessa [244], both dating back 4.0–3.6 myr. From the slightly younger (3.3–3.2 myr) Georgian site Kvabebi, *Vulpes* cf. *alopecoides* was described [245].

The evolutionary history of *V*. *vulpes* and *V*. *lagopus* started in the early Middle Pleistocene. The first appearances of the species are not older than 0.8 myr [235]. They are known from Stránská Skála (0.8–0.7 myr old) [145], L’Escale (0.7–0.6 myr old) [144], Bed 5 of Westbury-Sub-Mendip (0.6–0.5 myr old) [153], Arago Cave (0.6–0.5 myr old) [237,239], Vallparadís Estació (0.6–0.5 myr old) [237] and Hundsheim (0.6–0.5 myr old) [179]. Two chronosubspecies were recognised: *Vulpes vulpes jansoni* Bonifay 1971 from L’Escale (Bonifay 1971) and originally described as an independent species, *V. v. angustidens* (Thenius, 1954) from Hundsheim [179,240]. *V*. *v*. *jansoni* is characterised, when compared with the extant *V*. *vulpes*, by a more prominent protocone of the P4, M1 with a sharp paracone and metacone, strong lingual cingulum, distinct metaconule and hypocone, and M2 with a larger hypocone [144,237]. It was also concluded, however, that the main feature distinguishing *V*. *v*. *jansoni* from the extant *V*. *vulpes* is the development of the P4 protocone. Nevertheless, as shown in the broad revision of early *Vulpes* species, the shape and size of this cusp is highly variable in the extant *V*. *vulpes* [237]. Morphologically, it is indistinguishable from many extant specimens of the extant *V*. *vulpes* and falls within its range of variability. This resulted in this chronosubspecies no longer being valid [246].

*V*. *v*. *angustidens* was described as being slightly smaller than the extant *V*. *vulpes*, with an M1 with weaker protoconid, strong mesoconid and ridge-connected hypo- and entoconid, and narrow M2 with strong mesial cingulum, strong entoconid and metaconid higher than the protoconid [179]. In the revision of the European Middle Pleistocene, it was noted that similarly dated (0.6–0.5 myr old) *V*. *vulpes* [235] individuals, from the Vallparadís Section to that from Hundsheim, resemble them in the small size of the entoconid and the well-developed entoconulid on the M1 talonid. The material from Hundsheim was also personally studied by A.M. and we also found strong similarities with other Middle Pleistocene specimens [222]. From Poland, the earliest occurrences of *V*. *v*. *angustidens* from Południowa Cave and Rębielice Królewskie 2 are similarly dated like other European localities from 0.7–0.6 myr ago. Younger records are those from Tunel Wielki Cave (MIS 14-12) and Draby 3 (MIS 11). We found some small morphometrical differences between them and the extant *V*. *vulpes*, and on this background assigned them to *V*. *v*. *angustidens* [247]. However, as was noted by Szuma [248,249,250,251,252], the enormous geographic range of the extant *V*. *vulpes* and a number of factors associated therewith, such as the geographic, climatic and biological factors, latitude, habitat productivity, differential food availability, competition (on various levels, i.e., intraguild, intrafamily or intraspecific), character displacement, genetic diversity and population density, resulted in a high metrical and morphological variability [247]. On this term, a further revision of the Middle Pleistocene *V*. *vulpes* from Europe, based on a higher number of specimens, is required. Until then, we maintain a taxonomic determination of *V*. *v*. *angustidens* as a valid chronosubspecies.

The evolutionary history of *V*. *lagopus* is much less clear and is very fragmentary and patchy. Firstly, it was recorded in the Olyorian Fauna (Siberia) ca. 700 kyr ago [253,254] but later disappeared from the fossil record for a long time. There is a consistent hiatus in the European fossil history of *V*. *lagopus* during the Middle Pleistocene, since the oldest European records of this species are not older than MIS 6 [186,255]. During the Late Pleistocene (MIS 5d-2), it was a member of the widespread *Mammuthus*-*Coelodonta* faunal complex that ranged from Eastern Siberia to Spain [247,254,255].

In this context, Polish records of this species, especially those dated at MIS 9-7, are especially noteworthy. Specimens from the lowermost layers of the Biśnik Cave (19ad-19) and Deszczowa Cave (layers I-V), dated at MIS 9-8, showed morphology with an admixture of intermediate features between *V*. *lagopus* and *V*. *praeglacialis* [61,63]. The material from older sites, dated at MIS 15-13 such as Rębielice Królewskie 2 or Południowa Cave, is still classified as *V*. *praeglacialis*. The material dated as MIS 9 and younger was already determined as *V*. *lagopus*, but morphologically is still far away from the Late Pleistocene (MIS 5d-2) specimens. They are small and gracile like younger individuals but, simultaneously, still possesses a less curved lower margin of the mandibular body, proportionally longer diastemas between C1/c1 and P1/p1, and less reduced entoconid on the M1, and still present ridge-connected hypo- and entoconid on the M1 talonid. Later dated (MIS 7 or younger) specimens morphologically resembled the Late Pleistocene and the extant *V*. *lagopus*, and the distinguishing differences are minor [247]. Contrary to *V*. *vulpes*, for *V*. *lagopus,* the geographic and climatic factors were less important. Its morphology was influenced more by its interspecific competition with *V*. *vulpes* and food source variability and accessibility [248,249,250,251,252]. Different species are susceptible to different combinations of these proxies with different variability and, eventually, features [240,252,253,254]. It was also showed that these proxies more severely affected the morphology in the case of sympatry between different species of *Vulpes*, such as that between *V*. *vulpes* and *V*. *lagopus*, which accentuated the variability. Additionally, Szuma [248] found an important relationship between morphotypes and climatic parameters such as the air moisture and mean annual temperature. This may have a great relevance for paleontological studies; however, it is still poorly studied [247].

## 5. Conclusions

The analysis of Polish canids implies a strong geographical variation. The body size of canids demonstrates a clinal gradient correlated with climatic and environmental factors. This was well-documented and noted for the extant canids and the same can be expected for the extinct species. Body size cannot be strictly taken as a good indicator in taxonomy. However, it certainly constitutes important ecological characters in this process.

We summarised over 170 years of research on canids in Poland against the backdrop of Europe. Despite such a long period of research, relatively few publications on this subject were published during this time. In the case of many species, Poland is still a blank spot in biostratigraphic and palaeoecological analyses. However, as research has shown, this is usually not due to the fact that they do not exist in this area, but rather is because their remains have not been found as of yet. It often has to do with the lack of proper recognition of the already accumulated material. The extensive revision of canine material carried out consistently for many years in conjunction with new finds allowed for a significant expansion of the knowledge about their spatial and temporal occurrence in Poland. In biostratigraphic and palaeoecological analyses, the area of Poland that is particularly important to consider is its geographical location. Predominantly lowland or upland in the southern part, devoid of significant ecological barriers with the exception of three major rivers (Bug, Wisła, Odra), it has been and is an important migration corridor in the East–West system. This 500–600 km wide corridor, bounded in the north by the Baltic Sea and in the south by the arch of the Carpathians and Sudetes, was the Asian gateway to Europe, from where species of eastern origin penetrated into the interior of the continent. In colder periods, it was, in turn, a region through which boreal species or those associated with the mammoth steppe retreated.

Compared to Poland, areas such as Western Europe are a peripheral zone of Eurasia and migrations or evolutionary processes occurred in limited forms. Such peripheral zones are characterised rather by a greater variability that more reflects the biogeographical conditions. This is also an area better suited to possible survivals of more archaic subspecies or species in some isolated corners of Southern and Southwestern Europe, compared to the open, flat, vast lowlands of Central and Eastern Europe. This study confirms the importance of canids as biomarkers within most of the taxa considered likely in relation with climatic environmental changes.

The revision of the Polish canid fauna based on 111 sites, mainly cave and karstic localities, but also open-air sites, showed the presence of 12 canid species. These records covered a time span of the last 4.9 myr. Canid material, although found at numerous sites, is often represented by just a few or fragmentary finds. In many cases, these remains have a low taxonomic value due to a small number of poorly expressed diagnostic features. For this reason, the earliest history of the evolution of this family is relatively poorly documented in Poland and requires further research.

## Figures and Tables

**Figure 1 genes-14-00539-f001:**
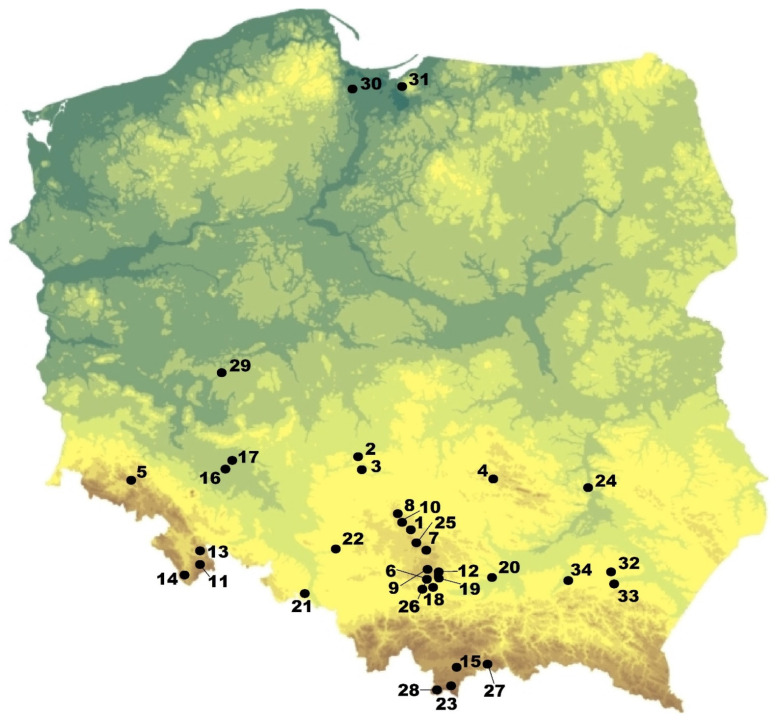
Location of paleontological sites with canid remains in Poland: 1—Podlesice, Podlesice Shelter, Żabia Cave and Babie Nogi Shelter, Zamkowa Dolna Cave, Kadzielnia, Kielniki, Przymiłowice, Towarna Cave, Olsztyńska i Wszystkich Świętych Cave and Ostrężnicka Cave; 2—Mała Cave, Węże 1, Ewy Cave, Raciszyn 1, Zalesiaki, Węże 2 and Draby; 3—Mokra (1 and 2), Rębielice Królewskie and Kamyk; 4—Kozi Grzbiet, Sitkówka, Raj Cave and Zygmuntówka Quarry; 5—Południowa Cave, Wschodnia Cave, Naciekowa Cave, Obok Wschodniej Cave, Północna Duża Cave, Cisowe Shelter (1 and 2), Małgorzaty Shelter, Kuny Shelter, Wilcze Shelter, Panna Shelter, Trwoga Paleontologa Shelter, Aven w Połomie Cave and Kryształowa Cave; 6—Tunel Wielki Cave, Sąspowska Zachodnia Cave, Bramka Shelter and Zalas Shelter; 7—Biśnik Cave, Zegar Cave, Mroczna w Pośrednicy Cave, Jasna Strzegowska Cave, Shelter in Pośrednica 2, Pośrednie Shelter and Jasna Smoleńska Cave; 8—Deszczowa Cave, Niedźwiedzia Górna Cave and Krucza Skała Shelter; 9—Wierzchowska Górna Cave, Nietoperzowa Cave, Mamutowa Cave, Koziarnia Cave, Łokietka Cave, Gorenicka Cave, Żytnia Skała Shelter, na Miłaszówce Cave, Murek Cave and Shelter above Niedostępna Cave; 10—Dziadowa Skała Cave, Stajnia Cave, Górne Shelter above Stajnia Cave, Cave no 4 on Mount Birów, Ruska Skała Shelter, Komarowa Cave and Wilcze 1 Shelter; 11—Niedźwiedzia Cave, Na Ścianie Cave, Biały Kamień Cave, Kontaktowa Cave, Rogóżka Cave and Small Przy Torach Cave; 12—Ciemna Cave, Zbójecka Cave, Zawalona Cave, Puchacza Skała Shelter and Duża Cave at Mączna Skała; 13—Radochowska Cave; 14—Solna Jama Cave; 15—Obłazowa Cave and Obłazowa 2; 16—Wrocław-Hallera; 17—Skarszyn; 18—Kraków-Spadzista, Borsuka Cave, na Gołąbcu Cave, Maszycka Cave, nad Galoską Cave and na Gaiku 2 Shelter; 19—Borsucza Cave and nad Mosurem Starym Duża Cave; 20—Jaksice 2; 21—Pietraszyn; 22—Pyskowice; 23—Magurska Cave; 24—Wilczyce; 25—Okiennik Cave; 26—Przegińska Cave; 27—in Sobczański Wąwóz Cave; 28—Poszukiwaczy Skarbów Cave, Ciasna Cave; 29—Krobia; 30—Tczew; 31—Elbląg; 32—Trzebowisko; 33—Rzeszów (Wisłoka river); 34—Wielopolka.

**Figure 2 genes-14-00539-f002:**
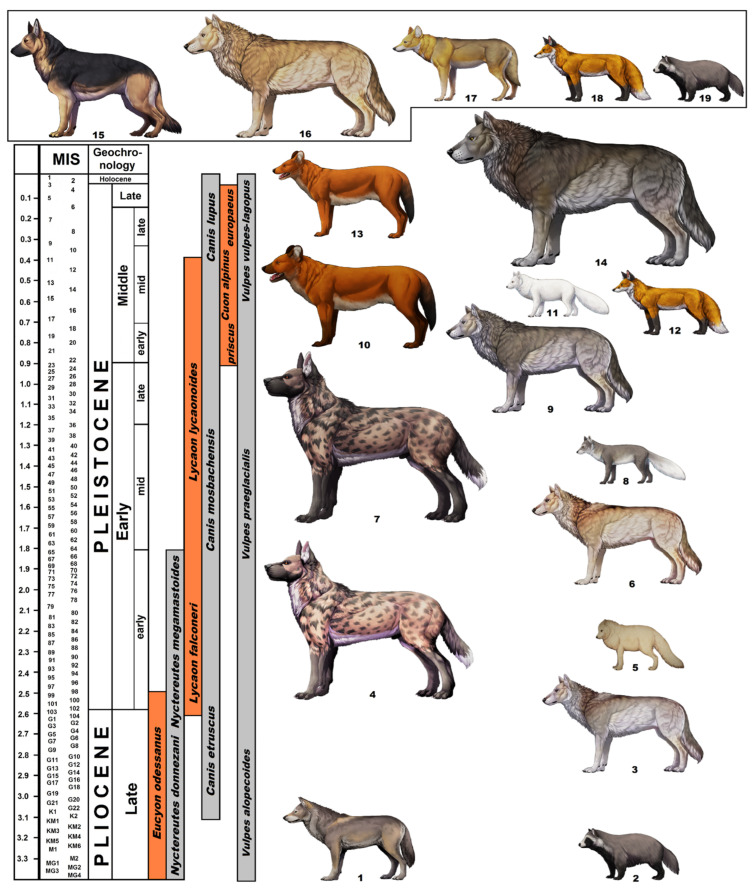
Biochronology of Polish canids (late Pliocene to Holocene): 1—*E. odessanus*, 2—*N. donnezani*, 3—*C. etruscus*, 4—*L. falconeri*, 5—*V. alopecoides*, 6—*C. mosbachensis*, 7—*L. lycaonoides*, 8—*V. praeglacialis*, 9—*C. l. lunellensis*, 10—*C. a. priscus*, 11—*V. lagopus*, 12—*V. vulpes*, 13—*C. a. europaeus*, 14—*C. l. spelaeus*. Extant, Polish canids are separated and showed in the rectangle: 15—*C. l. familiaris* (a big specimen of the common dog race German shepherd, with a withers height of 70 cm, for comparison to wild canids), 16—*C. l. lupus*, 17—*C. aureus*, 18—*V. vulpes*, 19—*N. procyonoides*. Artwork by W. Gornig.

**Figure 3 genes-14-00539-f003:**
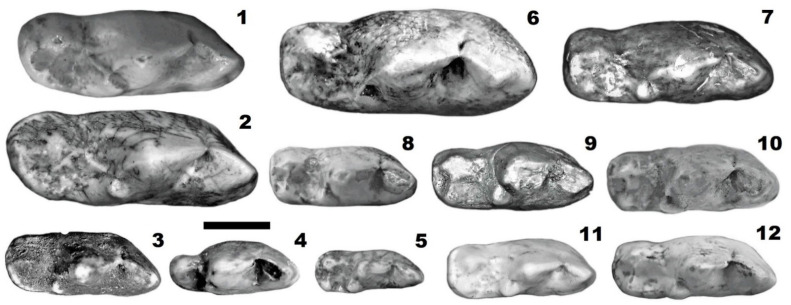
Lower carnassials (m1) of the Pleistocene canids from Poland: 1—*L. lycaonoides* from Żabia Cave (1.7–1.5 myr ago), 2—*L. lycaonoides* from Draby (MIS 11), 3—*C. a. fossilis* from Wschodnia Cave (MIS 8), 4—*C. a. europaeus* from Rogóżka Cave (MIS 3), 5—*Vulpes vulpes angustidens* from Rębielice Królewskie 2 (MIS 15-13), 6—*C. l. spelaeus* from Radochowska Cave (MIS 3), 7—*C. lupus* ssp. from of Dziadowa Skała Cave, layer 3 (MIS 5e), 8—*C. mosbachensis* from Żabia Cave (1.7–1.5 myr ago), 9—*C. mosbachensis* from Południowa Cave (MIS 19-17), 10—*C. l. lunellensis* from Biśnik Cave, layer 19ad (MIS 9), 11—*C. etruscus* from Węże 2 (2.9–2.6 myr ago), 12—*C. mosbachensis* from Sitkówka (MIS 15-13). All teeth showed in the same scale, in the occlusal view. Scale bar 10 mm.

**Table 1 genes-14-00539-t001:** List of the main Polish sites with canid remains in chronological order.

Site	Age	*E. odessanus*	*N. donnezani*	*L. falconeri*	*L. lycaonoides*	*Cuon alpinus priscus*	*Cuon alpinus fossilis*	*Cuon alpinus europaeus*	*Canis etruscus*	*Canis mosbachensis*	*Canis lupus lunellensis*	*Canis lupus spelaeus*	*Canis lupus lupus*/ssp.	*Vulpes alopecoides*	*Vulpes praeglacialis*	*Vulpes vulpes*	*Vulpes lagopus*
Węże 1	3.8–3.4 myr																
Węże 2	2.8–2.6 myr																
Rębielice Królewskie 1A	2.4–2.2 myr																
Zamkowa Dolna Cave A	2.4–2.2 myr																
Kielniki 3B	2.2–1.8 myr																
Kadzielnia	2.2–1.8 myr																
Kamyk	1.9–1.8 myr																
Żabia Cave	1.7–1.5 myr																
Kielniki 3A	1.5–1.3 myr																
Zalesiaki 1A	1.1–0.9 myr																
Kielniki 1	1.1–0.8 myr																
Zamkowa Dolna Cave C	1.0–0.8 myr																
Przymiłowice	MIS 22-17																
Kozi Grzbiet	MIS 19-17																
Sitkówka	MIS 19-17																
Południowa Cave	MIS 17-15																
Rębielice Królewskie 2	MIS 15-13																
Tunel Wielki Cave	MIS 14-12																
Draby 3	MIS 11																
Biśnik Cave, l. 19ad-19	MIS 9																
Deszczowa Cave, l. 1-5	MIS 9-8																
Wschodnia Cave	MIS 8-7																
Biśnik Cave, l. 18-13	MIS 7-5e																
Wierzchowska Górna Cave	MIS 7-1																
Nietoperzowa Cave	MIS 6-1																
Komarowa Cave	MIS 6-1																
Stajnia Cave	MIS 6-1																
Dziadowa Skała Cave	MIS 5e-1																
Mamutowa Cave	MIS 5-1																
Koziarnia Cave	MIS 5-1																
Niedźwiedzia Cave	MIS 5d-1																
Raj Cave	MIS 5-1																
Biśnik Cave, l. 12-1	MIS 5d-1																
Radochowska Cave	MIS 5d-1																
Skarszyn	MIS 3																
Kraków-Spadzista	MIS 3																
Late Pleistocene ^1^	MIS 5d-1																
Holocene ^2^	MIS 1																

^1^ Various Late Pleistocene Polish sites are characterised by the same canid composition: *C. lupus*, *V. vulpes* and *V. lagopus*. Those localities are: Towarna Cave, Ciemna Cave, Tczew, Elbląg, Malschwitz, Naciekowa Cave, Obok Wschodniej Cave, Północna Duża Cave, Wschodnia Cave, Rogóżka Cave, Obłazowa Cave, Bramka Shelter, Żytnia Skała Shelter, Ruska Skała Shelter, Cisowe 1 Shelter, Biały Kamień Cave, Miniaturka Cave, Na Ścianie Cave, Aven w Połomie Cave, Zbójecka Cave, Gorenicka Cave, Okiennik Cave, na Miłaszówce Cave, na Gołąbcu Cave, nad Galoską Cave, na Gaiku 2 Shelter, Obłazowa Skała 2, Murek Cave, Trzebowisko, Przegińska Cave, Poszukiwaczy Skarbów Cave, Solna Jama Cave, Magurska Cave, Rzeszów, Wisłoka river, Wielopolka, Krucza Skała Shelter, Wrocław-Hallera, Maszycka Cave, Zalas Shelter. In some, such as in Naciekowa Cave, Obok Wschodniej Cave, Północna Duża Cave, Wschodnia Cave, Rogóżka Cave and Skarszyn, a presence of very large, robust cave wolf *C. l. spelaeus*, a characteristic element of glacial faunas, has been detected. ^2^ From numerous Holocene Polish localities (Sąspowska Zachodnia Cave, in Sobczański Wąwóz Cave, Puchacza Skała Shelter, Cisowe 2 Shelter, Małgorzaty Shelter, Kuny Shelter, Panna Shelter, Trwoga Paleontologa Shelter, Niedostępną Cave, Ciasna Cave, Puchacza Skała Cave, Nad Mosurem Starym Duża Cave, Kontaktowa Cave, Small Przy Torach Cave, Duża Cave at Mączna Skała, Zbójecka Cave, Gorenicka Cave, Okiennik Cave, na Miłaszówce Cave, na Gołąbcu Cave, nad Galoską Cave, na Gaiku 2 Shelter, Murek Cave), the occurrence of two canids was documented: *C. lupus*, *V. vulpes*. Additionally, in most of these paleontological sites as well as from a few thousands of archaeological localities, *Canis lupus familiaris* Linnaeus, 1758 was documented. The oldest remains of a domestic dog from Poland are not older than 11–10 kyr BP and are much younger than other European finds. The result is that they change absolutely nothing in the research on canine biochronology and therefore they are not the subject of this study. The presence of this species is only signalled and listing all the sites is beyond the scope of this work.

## Data Availability

The authors declare that all the data supporting the findings of this study are available within the article or from the corresponding author upon reasonable request.

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
