# Peer review of "History of Polish Canidae (Carnivora, Mammalia) and Their Biochronological Implications on the Eurasian Background"

_genes, 2023, doi:10.3390/genes14030539_

Round 1

Reviewer 1 Report

All my comments are highlighted in the manuscript (in the attached file). Please, focus your interest on references, minor deficiencies  in English, and Xenocyon/Lycaon taxonomy ...

Author Response

Comments from the Editors and Reviewers

First of all, we want to express our warmest greetings for very valuable and informative comments and critical opinions, which were very useful and allow us to see weak part of the manuscript and were the base for its re-build. Many of these comments are very useful in general, and allow us to avoid in the future similar mistakes. Thank You very much dear Editors and Reviewers.

Comment no 1. It would be interesting if more pictures could be added, especially of exceptional specimens  such as the first occurrence of Eucyon and Nyctereutes or Vulpes corsac.

Answer no 1. All of those new records are subjects of separate articles which are now under review. Article presented here is a general overview on biochronology of Polish canids, showing their abundance and uniqueness. All those publications a broad revision of Polish fossil canids.

Comment no 2. The changing body sizes of the members of the canid family present in the course of the Pleistocene are mentioned a lot; it would be useful to add some metric data to show those fluctuations. An interesting metric could be the crown length of the lower carnassial (observed ranges, means) grouped by (sub)species and/or time period.

Answer no 2. Such changing of body size was showed on the relation between Lycaon, Canis and Cuon. Remains of other Polish canids, maybe with the exception of the genus Vulpes, are still to rare to show reliable data.

Comment no 3. In lines 536-539 that discuss very briefly the genetics of Pleistocene wolves and the domestication of the dog, the recent paper of Bergström et al. (2022) should be included. Suggested reference: Bergström, A., Stanton, D.W.G., Taron, U.H. et al. 2022. Grey wolf genomic history reveals a dual ancestry of dogs. Nature 607, 313–320 https://doi.org/10.1038/s41586-022-04824-92.12.0.0 2.12.0.0.

Answer no 3. The paper was add.

Comment no 4. The English of the manuscript can be improved.

Answer no 4. The English has been improved

Comment no 5. I did not find the appendix.

Answer no 5. It was attached to the in the original version, but anyway we add the appendix once more.

Reviewer 2 Report

The manuscript “History of Polish Canidae (Carnivora, Mammalia) and their biochronological implications on the Eurasian background” gives an interesting overview of the (co-)occurrence of the Canidae in Poland since the Pliocene. I have a few comments.

It would be interesting if more pictures could be added, especially of exceptional specimens  such as the first occurrence of Eucyon and Nyctereutes or Vulpes corsac.

The changing body sizes of the members of the canid family present in the course of the Pleistocene are mentioned a lot; it would be useful to add some metric data to show those fluctuations. An interesting metric could be the crown length of the lower carnassial (observed ranges, means) grouped by (sub)species and/or time period.

In lines 536-539 that discuss very briefly the genetics of Pleistocene wolves and the domestication of the dog, the recent paper of Bergström et al. (2022) should be included.

The English of the manuscript can be improved.

I did not find the appendix.

Suggested reference:

Bergström, A., Stanton, D.W.G., Taron, U.H. et al. 2022. Grey wolf genomic history reveals a dual ancestry of dogs. Nature 607, 313–320 https://doi.org/10.1038/s41586-022-04824-9

2.12.0.0 2.12.0.0

Author Response

Comment no 6. Since there is still discussion on the genus status of this large canid, it could be good to write why you are using generic name Lycaon.

Answer no 6. We use term proposed and broadly used by many authors e.g. Madurell-Malapeira, Baryshnikov etc. In the authors opinion, based and supported by visits in many collection, all those large canids represent one, evolutionary lineage. But in this article, which is a general overview, it is not place for such detail analysis. The detail discussion of the taxonomic position of the genus Lycaon (Xenocyon) was given in another, yet reviewed paper.

Comment no 7. (I) Please check that all references are relevant to the contents of the manuscript.

Answer no 7. All references were carefully checked and properly cited within the text. Also some very few typographic errors were corrected.

Comment no 8. (II) Any revisions to the manuscript should be marked up using the “Track Changes” function if you are using MS Word/LaTeX, such that any changes can be easily viewed by the editors and reviewers.

Answer no 8. It was done.

Comment no 9. (III) Please provide a cover letter to explain, point by point, the details of the revisions to the manuscript and your responses to the referees’ comments.

Answer no 9. A cover letter to explain, point by point, the details of the revisions to the manuscript and your responses to the referees’ comments was add.

Comment no 10. (IV) If you found it impossible to address certain comments in the review reports, please include an explanation in your appeal.

Answer no 10. It was done.

Round 2

Reviewer 2 Report

The authors have replied to the remarks I made on the first version of the manuscript. The revised manuscript can be accepted in its present form.